# Understanding External Influences on Target Detection and Classification Using Camera Trap Images and Machine Learning

**DOI:** 10.3390/s22145386

**Published:** 2022-07-19

**Authors:** Sally O. A. Westworth, Carl Chalmers, Paul Fergus, Steven N. Longmore, Alex K. Piel, Serge A. Wich

**Affiliations:** 1School of Biological and Environmental Sciences, Liverpool John Moores University, Byrom Street, Liverpool L3 3AF, UK; s_westworth@hotmail.co.uk; 2School of Computer Science and Mathematics, Liverpool John Moores University, Byrom Street, Liverpool L3 3AF, UK; c.chalmers@ljmu.ac.uk (C.C.); p.fergus@ljmu.ac.uk (P.F.); 3Astrophysics Research Institute, Liverpool John Moores University, Liverpool L3 3AF, UK; s.n.longmore@ljmu.ac.uk; 4Department of Anthropology, University College London, Taviton Street, London WC1H OBW, UK; a.piel@ucl.ac.uk

**Keywords:** citizen scientists, poacher, wildlife, biodiversity conservation, automation

## Abstract

Using machine learning (ML) to automate camera trap (CT) image processing is advantageous for time-sensitive applications. However, little is currently known about the factors influencing such processing. Here, we evaluate the influence of occlusion, distance, vegetation type, size class, height, subject orientation towards the CT, species, time-of-day, colour, and analyst performance on wildlife/human detection and classification in CT images from western Tanzania. Additionally, we compared the detection and classification performance of analyst and ML approaches. We obtained wildlife data through pre-existing CT images and human data using voluntary participants for CT experiments. We evaluated the analyst and ML approaches at the detection and classification level. Factors such as distance and occlusion, coupled with increased vegetation density, present the most significant effect on DP and CC. Overall, the results indicate a significantly higher detection probability (DP), 81.1%, and correct classification (CC) of 76.6% for the analyst approach when compared to ML which detected 41.1% and classified 47.5% of wildlife within CT images. However, both methods presented similar probabilities for daylight CT images, 69.4% (ML) and 71.8% (analysts), and dusk CT images, 17.6% (ML) and 16.2% (analysts), when detecting humans. Given that users carefully follow provided recommendations, we expect DP and CC to increase. In turn, the ML approach to CT image processing would be an excellent provision to support time-sensitive threat monitoring for biodiversity conservation.

## 1. Introduction

Monitoring biodiversity and its potential threats are prerequisites for successful biodiversity conservation. Such monitoring allows for documenting threats facing biodiversity and enables the evaluation of the impact of threats to biodiversity and the assessment of conservation strategies [1]. There are numerous and diverse methods to monitor biodiversity and the threats facing it. These have ranged from ground-truth data collection by field teams, to remotely sensed data collected from satellites, camera traps (CTs), autonomous acoustic recording units, and drones [2,3,4,5,6,7,8,9,10]. However, the ease of use, endurance capabilities, and low-cost maintenance of camera traps, as well as the easy to interpret data that they provide, make this a prevalent technology for wildlife monitoring and associated threats, including poaching, human-wildlife conflict, and the effects of habitat degradation [3,7,8,9]. For example, the average CT battery life enables continuous target monitoring, and even at high-sensitivity rates with increased CT image captures, batteries can last, on average, two to four weeks [10]. This is a considerable advantage compared to similar surveillance and monitoring tools, including UAVs with an average of 20–30 minutes flight time. Except for more advanced UAVs, including solar powered, reaching 25 days, this is not a widely utilised approach due to their current payload capacities and fragile components [11]. In addition, the advancement in automated networked CTs for image communications allows for more efficient data collection and transmission [9], without the need for full-time human operation as with UAV applications. Such advancements in CT image transmissions, coupled with long-term battery life, enable continuous, near-real-time data transmission. However, even with CT advancements, the vast number of images that often result from deployments pose a challenge to users. This hinders the application of CTs as a full near-real-time tool, as image analysts, who are usually experts but increasingly also citizen scientists (trained volunteers) [12], must assess images manually to detect, identify, and count targets, which is a time-consuming process [12,13,14,15]. This prevents the effective application of CTs in time-sensitive situations, such as monitoring threats including poachers, as it is unrealistic to expect analysts to be on standby for image evaluation, regardless of image transmission platforms, e.g., satellites or GSM links [16].

With the intention to increase CT image processing speeds for time-sensitive applications, some research groups have integrated machine learning (ML) with citizen scientists to automatically detect and identify humans/animals (targets) within CT images [16,17,18]. Several groups have taken a further step towards automation and predominantly or solely used ML for such tasks [19,20,21,22,23]. These studies have indicated overall detection accuracies ranging from 68–93% [20,22], illustrating the capacity for ML to detect targets within previously tested conditions. 

Ideally, we would independently apply ML for object detection in CT images irrespective of potential influences on detection probability (DP) and correct classification (CC). However, many of these studies have used images of whole animals to reduce occlusion or used single-species images and simplified conditions, likely leading to increased accuracy biases. This includes Norouzzadeh et al. [19] using single-species images, likely contributing to accuracies of 96.6% on a 3.2 million-image Snapshot Serengeti dataset from Tanzania. Additionally, Norouzzadeh et al. [22] further investigated ML applications using the Snapshot Serengeti dataset to reduce overall manual processing and ML model training. This time they manually cropped and segmented images containing whole animal bodies to increase animal identification. This likely led to their ML performance matching state-of-the-art citizen scientist accuracies for the 3.2 million image dataset with only 14,100 manual classification labels, reducing manual labelling effort by over 99.5%. Similarly, Yu et al. [21] cropped over 7000 camera trap images from Barro Colorado Island, Panama and Hoge Veluwe National Park, Netherlands. This resulted in an 82% accuracy of detection and classification of 18 mammalian species using ML methods. 

Although in some cases, combining citizen scientist and ML methods may be beneficial, such methods have not been practiced for target detection and classification within CT images [17,18]. However, Willi et al. [17] combined analyst and ML methods, using citizen scientists to pre-process images. They then tested CNN performance on empty image identification accuracy (91–98%) and species identification accuracies (88–92%) of all CT images, reducing analyst image processing time by 43%. Nevertheless, the need for researchers to simplify conditions or combine analyst and ML processing methods for sufficient accuracies underlines the importance of mitigating external influences on target DP and CC in CT images. 

There is, however, increasing promise in ML methods as an independent approach to CT image processing, as studies focus on improving ML performance in comparison with analysts [22]. An example of this includes Thangarasu and Kaliappan [23], comparing machine-learning and deep-learning model accuracies for species identification and finding >95% accuracy when detecting and classifying 19 mammalian species. Nonetheless, such performance testing has not yet incorporated an extensive amount of external influential factors.

Certain factors’ influence, including effective detection distance and species size on the effect of CT trigger probability, are well documented [24,25,26]. However, there is no published study focusing on their influence on the ability to detect targets within CT images, although distance and species size class have both been identified as reliable predictors of wildlife DP due to their influence on CT trigger probability [25,26]. The influence of wildlife occlusion on the DP of targets in CT images, particularly coupled with increasing distance, has been of interest. Willi et al. [17], for example, found that increased distance from the CT increased the occlusion probability of the animal due to vegetation between the animal and the CT. Similarly, studies have found that occluded species-specific characteristics significantly influenced wildlife CC negatively. For example, Yu et al. [21] found negative influences of occluded species-specific characteristics, including coloured patches, stripes, spots, and overall body composition, on the CC of ungulates. Moreover, Gomez Villa et al. [16] found negative influences of partially occluded species-specific characteristics, including spots and antler shape, on the CC of red brocket and white-tailed deer species. 

Some studies have highlighted differences in light illumination and their influence on wildlife DP and CC in CT images, including Yu et al. [21] finding that dark lighting (dusk) presented a negative effect on DP. This also caused classification biases of diurnal species due to the appearance of dense foliage covering the CT, commonly leading to grayscale images. They also found biases of some cathemeral species frequently active during daylight. This is because, as expected, the authors did not train the ML algorithm to detect and classify diurnal species in grayscale images, and cathemeral species that are commonly active during daylight naturally lack grayscale images for model training. The biases caused such species to be commonly classified as cathemeral species, including *Crocuta crocuta.* Nonetheless, this study still resulted in 88.9% accuracy, using deep regions with convolutional neural networks (R-CNNs) to detect and correctly classify species in CT images from the Snapshot Serengeti dataset. 

No known study has focused on the influences of human-related factors and their effect on human DP and CC in CT images. That said, Hambrecht et al. [2] used thermal infrared (TIR) and red, green, and blue (RGB) drone imagery. They found human occlusion due to canopy density, increasing distance from the centreline—the centre of an experiment created by the investigator, and analyst performance were the main influences of human DP within TIR drone imagery. The authors also assessed clothing colour contrast relative to the background and its influence on human detection, finding that red and blue were the most influential colours for detecting humans using RGB drone imagery. 

Although many studies focus on varying factors and their influence on wildlife detection and classification in CT images, no study has focused explicitly on the potential influence of external factors on the DP and CC of wildlife and humans in CT images. Therefore, in this study, we assessed the potential influence of the following factors: species (characteristics), analyst performance, vegetation type, degree of occlusion, distance from the CT, time-of-day, human/wildlife (target) height (metres), colour contrasts of clothing, and orientation towards the CT on the DP and CC of targets within CT images (see Figure 1 for a visual overview of the tested factors within CT images). Additionally, we investigated multi-factor influences on ML performance compared with analyst performance for true-positive detection and classifications. We collected the data during the standard operating of the cameras within western Tanzania. Afterwards, we collated the training and testing data to assess the influence of the tested factors on target DP and CC within CT images, using ML and analyst methods. We then compared the performance of these methods for true-positive detections and classifications. 

Illustration of Camera Trap (CT) Image Scenarios Incorporating Tested Factors

**Figure 1 sensors-22-05386-f001:**
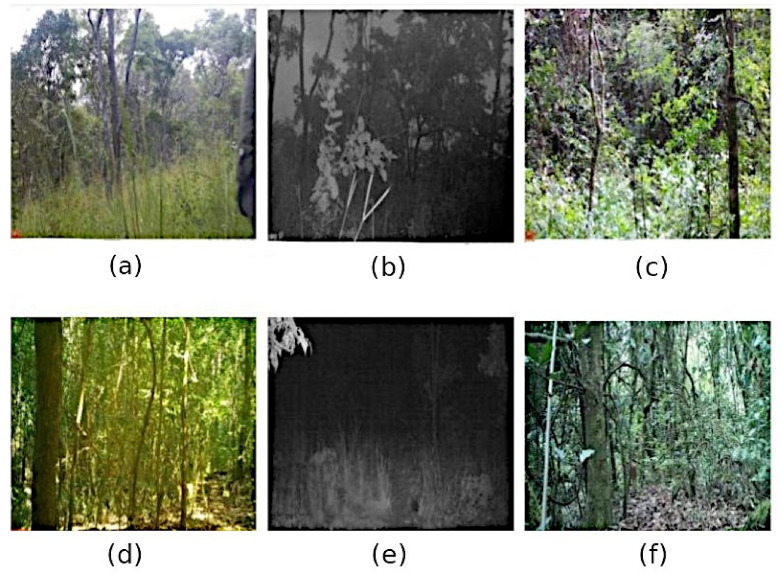
Examples illustrating different CT image scenarios, incorporating the factors measured in this study. Human images include (**a**) vegetation type (miombo woodland), time-of-day (daylight), distance (<5 m), the colour of clothing (green), orientation towards the CT (yes), height (1.9 m), and occlusion (68–100%); (**b**) Vegetation type (miombo woodland), time-of-day (dusk), distance (15–30 m), height (variations) and occlusion (68–100%); (**c**) Vegetation type (riverine forest), time-of-day (daylight), distance (10 m), the colour of clothing (green), orientation towards the CT (no), height (1.7 m), and occlusion (68–100%). Wildlife image examples include (**d**) Species (*Philantomba monticola*), vegetation type (miombo woodland), time-of-day (daylight), distance (0–4.9 m), orientation towards the CT (no), and occlusion (68–100%); (**e**) Species (*Crocuta crocuta*), vegetation type (miombo woodland), time-of-day (dusk), distance (5–9.9 m), orientation towards the CT (no), and occlusion (34–67%); (**f**) Species (*Tragelaphus sylvaticus*), vegetation type (riverine forest), time-of-day (daylight), distance (>10 m), orientation towards the CT (no), and occlusion (68–100%).

We initially proposed an alternative hypothesis for each factor tested. This includes: A human or animal’s increase in distance and occlusion simultaneously from the CT would negatively influence at least one experiment, and denser vegetation cover would further contribute to this influence for one or more models.A darker (green) colour contrast in comparison to the background would negatively influence human DP.Decreases in the size of humans or wildlife would decrease their probability of detection, respectively, and darker times of day (dusk) would negatively affect human or wildlife DP and CC.Occluded or partially occluded characteristic traits for similar species would decrease wildlife CC.DP and CC would increase with increased human or animal orientation towards the CT.There would be a significant positive difference in analyst performance on target DP and CC.Lastly, the ML method would present higher or equivalent true-positive detections and classifications than the analyst method for one or more experimental models.

To test our hypotheses and meet the study’s aims, we have structured this article as follows: in Section 2, we provide a detailed description of the experimental setup and analytical processes. Section 3 provides quantitative data results, presenting the degree of influence of the analysed factors and comparing the performance of both ML and analyst methods. This finally leads to Section 4, which reviews the impact of the tested factors, concluding with several recommendations and discussing the potential of ML to detect and correctly classify targets within CT images irrespective of such influencing factors.

## 2. Materials and Methods

### 2.1. Study Site

We collected data at the Issa Valley research station in the Tongwe West Forest Reserve in western Tanzania. The site is located approximately 1500 m above sea level. The dominating vegetation within the surrounding area is miombo woodland (*Brachystegia* and *Julbernardia*), interspersed with a mixture of open (rocky outcrops and grasslands) and closed (riverine forest) vegetation [27]. 

### 2.2. Experimental Design for Human CT Images

We conducted the human CT experiments between 13–19 March 2020 (see Institutional Review Board Statement). We used two Bushnell Trophy Cam HD Aggressor 2017 (model number: 119877, Bushnell, Overland Park, KS, USA) CTs to collect data. We also varied the following factors during the experiments: clothing colour, time-of-day, the distance between the human and CT, occlusion, the orientation of the human relative to the CT, and human height, with naturally varying tree density. A total of 15 human participants were involved, a mixture of LJMU students and local staff, who provided prior and informed consent. Each experiment varied in participant count, between two and six, depending on availability. Both green and blue coats and trousers were used interchangeably by each participant during the experiment, resulting in a completely blue and green set of clothing for each participant for each time-of-day, condition, and distance. There was a total of 11 daylight experiments (between 7:15–8:00) and 11 dusk experiments (between 19:15–20:00) within the miombo woodland (low tree density). The riverine forest (high tree density) experiments were conducted consecutively on 19 March 2020 (between 10:00–15:00), resulting in six daylight experiments. 

For the daylight experiments, participants walked from right-left horizontally (relative to the CT) across the 5-m marker. Once participants reached the end, they walked up the side of the experiment field (out of the CTs field of view) to the left-hand side of the 10-m marker and walked left-right across the 10-m distance. They continued this pattern across each set distance, starting at 5 m up to 30 m; once completed, participants returned to the centre of the 30-m distance marker and walked through the centre, directly towards the CT (see Appendix B Figure A1 for a visual representation). For the dusk experiments, we randomly allocated participants a specific distance to walk across to the end and return to their starting position (see Appendix B Figure A2 for a visual representation). 

We used a 5-m tape measure to measure the distance from the CT using 5-m intervals (<5, 5, 10, 15, 20, 25, and 30 m) and marked each distance using a fallen branch (central to the CT). The <5 m class included all images taken of participants, between the CT and 5-m distance marker, excluding dusk CT experiments. Next, we measured the CTs field of view by using a live feed video and having an individual walk across each set distance marker and stop once their body left the side of the CTs field of view; the individual then placed a stick on the ground as the perimeter marker. We did this at both ends of each set distance marker. Afterwards, we measured the distance between the left and right markers for each distance class and used these measurements as reference points for additional experiments. Proceeding this, we replaced the fallen branch marking each distance class with yarn, tying it to the left and right perimeter markers, repeating for each distance class, enabling the participants to follow the yarn to the end of the marker. We then tested participant height variations, ranging between 1.5–1.9 m, using a 5-m tape measure. 

Once we captured the footage, we first measured target occlusion from the CT in classes (0–33%, 34–67%, and 68–100%). We used the following rule to identify occlusion—target occlusion occurs if any part of the body is not visible in the image, either due to obstruction of an object or the CTs field of view. We proceeded to measure the factor orientation towards the CT by selecting footage of the target moving or facing in the CTs direction. We then selected and organised the footage of humans orientated towards the CT into each set distance marker category. This included selecting footage captured at the following distance markers (<5, 5, 10, 15, 20, 25, and 30 m). We also associated all measured factors with their appropriate classification labels (see Table 1 for classification label overview). Using Microsoft Excel Office 365 version 1.0.1, Microsoft Corporation, Washington, DC, USA, we allocated a column for each factor classification label. Each row was identified as a factor combination captured within a CT image, as explained previously. Finally, we numerically labelled all individual experiments (experiment number) to account for potential influences between experiments within each model.

### 2.3. Wildlife Image CT Data

GMERC Ltd. (Busongola, Tanzania) researchers previously obtained the wildlife CT data at the Issa research station between 2016–2020 from Bushnell fixed-focal-length CTs. We initially selected 15 species, 4 of which were removed during the model training, resulting in 11 species for testing. We selected the following species *C. crocuta*, *Philantomba monticola*, *Potamochoerus larvatus*, *Tragelaphus sylvaticus*, *Genetta tigrina*, *Pan troglodytes schweinfurthii*, *Sylvicapra grimmia*, *Hystrix cristata*, *Panthera pardus*, *Cercopithecus ascanius*, and *Papio cynocephalus*. Following this, we selected the factors measured depending on their frequency within the species-specific CT footage and based on previous findings [16,17,19,20,21,22,23]. 

The chosen factors include each species’ height (shoulder height), species differences (species-specific visual traits), time-of-day (for cathemeral species), the distance of the animal to the CT, occlusion of an animal from the CT, and animal orientation relative to the CT, as explained below. We identified each species’ shoulder heights, previously assessed using the perpendicular distance between the base of the heel and the shoulder blade as according to Kingdon [28] and categorised these measurements into three categories (small 0–4 m, medium 4.1–7 m, and large 7.1–10 m). We used the factor species to represent species ID and general species-specific characteristics, including colour, body shape, and antlers/horns. Proceeding this, we identified each species’ circadian cycles, including cathemeral (daylight/dusk), nocturnal (dusk), and diurnal (daylight) [29,30,31,32,33,34,35,36,37]. This enabled us to ensure adequate footage was obtained of cathemeral species at both daylight (7:00–18:00) and dusk (18:01–5:00) for measuring the time-of-day factor. We measured the distance factor in classes (0–5, 5–10, and >10 m) at five different tree tag locations (4800, 3219, 4817, 4816, and 4815), selected based on the level of species abundant at each site. Once we had set the distance classes, we recorded the distance and position (relative to the CT) of the most predominant trees within each class in a forward-facing direction to use as reference points for measuring species distances within pre-captured CT images. GMERC Ltd. researchers captured all CT footage with a fixed focal length lens, using consistent video resolution (1900 × 1080). This allowed us to apply the distance measurements (explained above) to all CT footage. We measured the factor’s occlusion and orientation towards the CT identical to the human experiments (as described above). However, for wildlife experiments, we selected and organised all the footage containing wildlife orientating towards the CT into categories depending on their distance (0–5, 5–10, >10 m). As in the human experiment, we classified and organised the factors into their associated classification labels (see Table 1 for classification label overview). Each column represented a factor, and each row represented a factor combination captured within the CT image. Finally, we used the recorded tree tag number associated with each CT video sequence to account for potential differences between experimental locations.

### 2.4. Data Analysis

#### 2.4.1. CT Data Preparation

We filtered all video footage to remove unnecessary repeats, blanks (without target), and blurred footage. We determined the removal of footage by the following rules: remove consecutive frames of repeated scenes, keeping a maximum of four per sequence, and remove all empty footage/frames (without targets). We then converted the footage into a raw.jpg image sequences through a video conversion software VLC media player https://www.videolan.org/ (accessed on 3 May 2020), extracting one–four frames per second (fps) using the scene filter preference, with a consistent 1900 × 1080 image resolution. 

#### 2.4.2. Training and Testing ML and Analyst Methods

We used the ConservationAI website www.conservationai.co.uk (accessed on 5 December 2019) for ML training and testing using a Faster R-CNN Residual Neural Network (ResNet) 101 Common Objects in Context (COCO) model. We also used transfer learning to reduce the required training datasets. This was performed through the frozen weights of the ResNet101 model and we retrained the fully connected multilayer perceptron (MLP)—a multi-layer neural network connecting each layer in a one-way directed graph. We used the MLP supervised learning technique as it increases model fine-tuning through backpropagation. We trained the model (using the Visual Object Tagging Tool (VoTT) version 1.7.0, Microsoft Corporation, Washington, DC, USA), with a threshold of 1000 images per class to balance all classes, preventing classification bias within the model.

Firstly, we extracted the human training dataset from the experimental footage and collated the wildlife training dataset using images from the extracted CT footage, iNaturalist, and the Endangered Wildlife Trust (EWT). Secondly, we used between 100–200 images for each factor combination for both human and wildlife training datasets. Lastly, we sorted the wildlife (16,333 images) and human (11,781 images) testing datasets into four copies, one for the ML model and three for the analysts (citizen scientists) (see Appendix C Table A1 for training and testing dataset counts per class). 

Once we trained the automated model on the training dataset, we uploaded the testing dataset to the ConservationAI website. The algorithm then processed all images for wildlife and human detection and classification instances. For analyst (citizen scientist) training, we obtained three images per species from iNaturalist. We then distributed them to the three analysts for two days to memorise their features and associated classification labels. Following this, we provided the testing datasets in a random order to prevent pattern recognition and inference of the target location within the image based on a previous location. We used the Google images app, version 5.7.0.327437691, Microsoft Corporation, Washington, DC, USA, for the analyst detection and classification counts. The process involved drawing a bounding box over the target’s parameters—the outer edge of the object’s body, and typing the proposed classification name within the attached comments box, as explained below. 

After analyst and ML detection, we processed the images, resulting in the following potential options: (1) false-negative detection and no classification, (2) true-positive detection and false-negative classification, or (3) true-positive detection and classification (see Table 2 and Table 3 for detection and classification performance results). The definition of such options is as follows—a bounding box surrounding the target’s parameters would be a true-positive detection; the correct labelling/classification of the detected target would be a true-positive detection and classification; if there is no bounding box or it is not surrounding the target’s parameters, it is a false-negative detection and no classification; if the target is detected but mislabelled as another class, it is a true-positive detection and false-negative classification. 

We proceeded to categorise datasets in Microsoft Excel Office 365 version 1.0.1 into combinations of the variables included within the image and sorted them into their specific models. We gave the CC variable a not applicable (na) value in the event of a false-negative detection response, and we stacked the analyst data with corresponding number labels to analyse the three analyst response results simultaneously, assessing between analyst differences.

#### 2.4.3. Wildlife and Human Experimental Factor Analysis

We used a linear mixed model (glmm) with a logit link function, using the glmer function within the lme4 package in Rstudio version 3.6.3, as Winter [38] recommended. We first converted all categorical variables to factors with associated levels (see Table 1 for factor classes and their levels) using the “as.factor” function, version 3.6.2. We then excluded data entries with missing values using the “na.omit” function, version 0.10.6. For the CC analysis, this included 16,343 of 19,842 ML-dusk entries and 2590 of 8474 ML-daylight entries from the human datasets as well as 9272 of 39,708 analyst entries and 3341 of 7037 ML entries from the wildlife datasets. We then mean-centred all continuous variables and used the “all_fit” function from the “afex” package, version 0.27–2 [39], to mitigate convergence issues.

#### 2.4.4. Tested Variables within the Human and Wildlife Experiments

We produced a baseline model of all fixed effects and the associated random effect for the human and wildlife experiments for all detection and classification models. An example of the wildlife DP model baseline formula is: (DP outcome ~ distance + occlusion + time-of-day + species + orientation towards the CT + size class + analyst performance + (tree tag number|1)). To ensure the accuracy of the model, we tested all model assumptions using the package “performance”, function “check_model”, version 0.4.8 [40]. The performance test showed high multicollinearity between the size class and species variable. Therefore, we removed the size class variable to use the most explanatory variable (species), as Kim [41] recommended.

We created models for each category and ranked them using the “performance” package based on their Akaike Information Criterion (AIC) (lower AIC illustrates a better model fit), as advised by Winter [38]. The best-fit models of each category in Section 3.1 and Section 3.2, composed of forest plots, contained exponentiated odds ratios as effect sizes and corresponding 95 percent (%) confidence intervals as error bars. We also included the standard deviation (SD) for the random effect to illustrate the variance between groups (experiment locations/data collection locations). Additionally, we used the baseline variable as a comparison for each corresponding variable as defined in Section 3.1 and Section 3.2. 

#### 2.4.5. Outlier Sensitivity Analysis

We conducted a sensitivity analysis on the outliers from the human models, as mentioned below, to test their influence on their corresponding models and determine whether to remove them. We reanalysed the same data identical to the previous analysis after extracting the outliers and presented pre-outlier and post-outlier models for each instance. Firstly, we removed the 5-m distance variables from the analyst-daylight-DP model (see Appendix A) and the ML-daylight-DP model (see Appendix A). Secondly, we removed the distance variables 25 and 30 m from the ML-dusk-DP model (see Appendix A). Lastly, we removed the 10-m distance variable from the ML-dusk-CC model (see Appendix A). Upon completion of the sensitivity analysis, we chose the original data, including outliers, as we were confident that the outliers occurred due to natural conditions within the experiments. We determined this due to the following outcomes: the ML approach did not detect any humans at 25 and 30 m in dusk CT images, and both approaches detected and classified all humans correctly at a 5 m distance in daylight CT images. 

#### 2.4.6. ML and Analyst Performance Analysis

We extracted overall percentages for the three potential outcomes (1) true-positive detection and classification, (2) true-positive detection and false-negative classification, and (3) false-negative detection and no classification, for both ML and analyst methods. Moreover, we used a two-tailed z-test to determine the overall significant difference in the percentage of detection probabilities and CCs between the ML and analyst methods for both experiments (see Table 2 for the wildlife experiment and Table 3 for the human experiment). 

The ML model performance was further tested using the multi-class confusion matrix classification method (see Appendix D Table A2 for a full overview of all tested classes), as advised by Mohajon [42]. We also derived the primary performance measures of the ML model for each tested class (see Appendix D Table A3 for an overview of ML performance per class). 

## 3. Results

### 3.1. Assessing External Factor Influences on Wildlife DP and CC in CT Images 

The glmm analysis showed that increasing distance (>10 m) and occlusion (68–100%) levels showed significant (*p* < 0.001) substantial adverse effects on wildlife DP (see Figure 2a,b), except for 5–9.9 m which significantly (*p* < 0.001) positively affected wildlife CC using ML methods (see Figure 2c), indicating that increasing distance and occlusion led to decreases in wildlife DP within CT images.

*P. larvatus*, *S. grimmia*, *H. cristata*, *T. sylvaticus,* and *C. crocuta* species significantly (*p* < 0.001) positively influenced analyst-DP (see Figure 2a). Except for *H. cristata*, all species were within the medium-large class, indicating that increasing size increased wildlife DP within CT images using analyst methods. However, *P. monticola* and *P. pardus* illustrated a non-significant (*p* = 0.710, 0.903) influence on wildlife DP in CT images using analysts (see Figure 2a). Additionally, all species classes significantly (*p* < 0.001) positively influenced ML-DP (see Figure 2b). 

On the other hand, most species classes at increasing distance and occlusion influenced CC. *P. monticola*, *T. sylvaticus*, *S. grimmia* (*p* < 0.001), *G. tigrine* (*p* = 0.001), and *C. crocuta* (*p* = 0.002) illustrated a significant positive influence on ML-CC. Yet, *C. ascanius* significantly negatively impacted wildlife CC in CT images using both ML (*p* = 0.007) and analyst (*p* < 0.001) approaches. However, *P. larvatus* (*p* = 0.018), *P. schweinfurthii* (*p* = 0.739), *H. cristata* (*p =* 0.687), and *P. pardus* (*p* = 0.224) illustrated a non-significant influence on ML-CC (see Figure 2c,d). The negative influence of occlusion coupled with species characteristics on CC is anticipated when including increasing distance and occlusion levels between 34–100%, resulting in the partial or complete occlusion of visible species-specific traits. 

The analyst performance factor showed that Analyst 1 presented the most significant (*p* < 0.001) negative influence on wildlife CC compared to Analyst 2 (*p* = 0.252), illustrating a non-significant influence. This indicates that individual abilities to recall visual features and their associated classification labels caused fluctuations in wildlife CC (see Figure 2c). 

Furthermore, the time-of-day factor (dusk) significantly (*p* < 0.001) negatively affected the DP and CC of wildlife in CT images through the analyst approach and DP through the ML approach (see Figure 2a,b,d). This result has highlighted the influence of darker conditions on decreased DP and CC of wildlife. In addition, time-of-day presented a non-significant (*p* = 0.039) influence on wildlife CC using ML methods (see Figure 2c).

Nevertheless, the orientation toward the CT factor illustrated a highly significant (*p* < 0.001) positive effect on the DP of wildlife in CT images using both analyst and ML methods (see Figure 2a,b). This factor presented the most significant favourable influence on DP for both analyst and ML methods. 

In addition, the random factor representing variance between CT image locations showed the highest SD for the analyst-DP model (SD 2.529), followed by ML-CC (SD 2.519), ML-DP (SD 1.237), and the analyst-CC (SD 0.849) models. 

### 3.2. Comparison of Analyst and ML Model Performance for Wildlife DP and CC

The three analysts detected 81.1% and correctly classified 76.6% of wildlife within CT images, whereas the ML method detected 41.1% and correctly classified 47.5% of all wildlife within CT images. A two-sample z-test showed a significant difference in the percentage of detections (*p* < 0.001), 95% confidence interval (0.381 and 0.397) and classifications (*p* < 0.001), 95% confidence interval (0.279 and 0.304) between the two approaches. Moreover, the ML and analyst methods illustrated higher false-negative detections and classifications than true-positive detections and classifications (see Table 2 for a comparison of ML and analyst detections and classifications). 

### 3.3. Analysing External Factor Influences on Human DP and CC

Human detection within daylight CT images was highly significant (*p* < 0.001) between 10–30 m with a negative influence on DP and CC, increasing as distance increased (see Figure 3a,b). Such positive influences of shorter distances on DP and CC decreased for dusk experiments using both ML and analyst methods and ML-CC (see Figure 3d,e). This highlights the difficulty of detecting and classifying humans in CT images under such conditions. 

Occlusion of 34–67% showed a highly significant (*p* < 0.001) positive influence on human DP in dusk CT images using analysts (see Figure 3c), potentially due to open vegetation, increasing the DP of humans in CT images even at medium occlusion levels in dusk lighting conditions. However, increases in distance and occlusion significantly decreased human DPs in daylight CT images using ML methods and dusk CT images using analyst methods (see Figure 3a,d). This was increasingly true when combined with dense vegetation (riverine forest), causing a steep decline of human DP in daylight CT images at shorter distances (15 m) for both ML and analyst methods (see Figure 3a,d). Moreover, the model shows that detecting and classifying humans at dusk in open vegetation (miombo woodland) from the same distance caused more positive effects than detecting them in daylight within denser vegetation (riverine forest) (see Figure 3b,e). That said, vegetation type (*p* = 0.981) and occlusion 34–67% and 68–100% (*p* = 0.933, 0.673), colour and distance 5–30 m (*p* = 0.034, 0.038, 0.027, 0.094, 0.164, and 0.387) presented non-significant effects on the ML-daylight-CC model (see Figure 3e). Moreover, 5 m distance presents non-significant effects on the ML (*p* = 0.308) and analyst-daylight-DP (*p* = 0.096) models (see Figure 3a,d). Occlusion of 34–67% and 68–100% also had a non-significant influence on the analyst-daylight-DP (*p* = 0.155, 0.095) and ML-dusk-CC models (*p* = 0.704, 0.995). Targets at a 10-m distance had a non-significant (*p* = 0.923) effect on the ML-dusk-CC model (see Figure 3a,f). Furthermore, the ample amount of non-significant influential factors on human DP is likely due to the influence of naturally occurring outliers and the high rate of detections in daylight CT images. Additionally, the overall abundance of CCs for humans in both daylight and dusk CT images likely influenced the degree of non-significant influential factors for the CC models.

On the other hand, darker contrast colours (green) significantly (*p* < 0.001) negatively affected human DP in daylight CT images, using analyst methods (see Figure 3a), and presented insignificant effects on human CC in dusk CT images using ML methods (see Figure 3c). This result leads to the assumption that clothing colour contrast against the background mostly influenced the analyst’s ability to detect humans within daylight CT images.

However, human orientation towards the CT presented a significant (*p* < 0.001) but weak positive influence on the DP of humans within daylight CT images, using analyst methods (see Figure 3a). This indicates that increased surface area relative to the CT was a significant predictor of human DP in daylight CT images. 

The models displayed significant variations between experimental locations, with the ML-dusk-CC model (SD 2.683) displaying the highest variation. This is followed by the analyst-dusk-DP model (SD 1.038), ML-daylight-DP model (SD 0.806), analyst-daylight-DP model (SD 0.799), ML-dusk-DP model (SD 0.714), and ML-daylight-CC model (SD 0.326), respectively. 

### 3.4. Comparison of Analyst and ML Model Performance on Human DP and CC

The analyst method detected 71.8% of humans within daylight CT images and 16.2% of humans within dusk CT images. Similarly, the ML method detected 69.4% of humans in daylight CT images and 17.6% of humans in dusk CT images. A two-sample z-test showed a significant difference in the percentage of detections for daylight (*p* < 0.001), 95% confidence interval (0.012, 0.034), but a non-significant difference for dusk (*p* = 0.149), 95% confidence interval (−0.010, 0.001) models. The ML-daylight models presented higher true-positive detection and classification rates than the other models, with the analyst-dusk model illustrating the highest false-negative detection rate (see Table 3 for a comparison of detection and classification responses for both methods).

## 4. Discussion

This study aimed to identify factors influencing target DP and CC in CT images and develop recommendations to mitigate such influences. We achieved this by using both ML and human analyst approaches, alongside performance comparisons of ML and analysts for target DP and CC under variable conditions. 

Given the results of the experiments, our theory is that specific factors do indeed significantly influence the detection and classification of targets using both ML and analysts. More specifically, factors of distance and occlusion, particularly when coupled with increased vegetation density, presented the most significant effect on DP and CC. There are various studies that have highlighted different influences on the detection or classification of targets within CT images using ML approaches [16,17,19,20,21,22,23]. However, none have explicitly evaluated all factors within this paper and their interacting effects on DP and CC within natural experimental conditions. Given that users consider the following recommendations and exercise the correct setup and design methods, the knowledge provided should facilitate increased DPs and CCs of wildlife and humans within CT images using both ML and analyst approaches. Therefore, this paper contributes toward the successful application of ML methods as a tool for target DP and CC within CT images, for the time-sensitive monitoring of threats facing biodiversity, regardless of external influences. 

As an equation, the main theory would be as follows:A = Distance + Occlusion + Vegetation Type,
B = Analyst Performance + Orientation Towards the CT
where A is significantly reduced detection and classification of targets for most tested models and B is the most significant increase in DP and CC of targets. 

### 4.1. Important Factors to Consider for Improving Target DP and CC

The reduction in effective detection and classification distance of targets in CT images with increased distances (10–30 m) is comparable with previous studies. This includes Norouzzadeh et al. [22] who found reduced effective detection distance when assessing the DP of species with varying body mass on CT trigger probability. This is consistent with Findlay et al. [43] who noted that close (1 m) and far passes towards the peripheral viewpoint of the CT decreased trigger probability. Similarly, Hambrecht et al. [2] reported a significant influence of distance from the centreline on the DP of ‘poachers’ using drones and human analysts. Additionally, Marin et al. [44] reported the complexities of ML detection and classification of ‘poachers’ in partially occluded images. Likewise, the impact of increased occlusion levels on detection and classification within most experimental conditions agrees with previous findings [16,21].

When combined with further distances and occlusion, closed vegetation (riverine forest) led to increased adverse effects of distance and occlusion on human DP in all daylight models. The closed vegetation presented a more significant negative influence on detecting and classifying humans in daylight CT images than within dusk CT images captured in open vegetation (miombo woodland) at the same distance and occlusion. Similarly, Bukombe et al. [26] reported that although seasonal differences alone presented no influence, they adversely reduced the DP of ungulates in CT images when coupled with species size and increased distance. However, we only compared vegetation type differences for human daylight CT experiments. Therefore, we could not rule out denser vegetation as an influence on target DP and CC within dusk CT images. Similarly, vegetation type was not included in the wildlife experiments as only images containing riverine forest vegetation were used due to a lack of wildlife images captured in open woodland.

Three main factors contribute to the magnitude of non-significant effects on human CC and DP models: (1) the natural outlier conditions found at a 5 m distance for daylight DP models and 25–30 m distance for the ML-dusk-DP model, (2) the high probability that colour presents no influence on the CC of humans, and (3) the high true-positive detection and classification responses increase the likelihood that most factors presented a positive influence or no influence at all on DP or CC rates. This is because analysts presented no false-negative responses, and they reached 71.7% for daylight true-positive DPs, similar to ML with 76.5%. 

Dusk experiments presented a significant adverse effect on wildlife DP and CC for cathemeral species. This strongly supports the conclusions of Gomez Villa et al. [16] who found that greyscale CT images likely caused diurnal and cathemeral species classification biases towards commonly nocturnal species, including *C. crocuta*. Similarly, dusk lighting conditions significantly affected the DP of humans, particularly at further distances (20–30 m). However, Hambrecht et al. [2] found the time-of-day factor to be an insignificant predictor of human DP in drone imagery. 

Most of the species’ classes influenced DP and CC, with all but *C. ascanius* significantly negatively impacting wildlife CC in CT images using both ML and analyst approaches. This was particularly the case when correctly classifying species of similar characteristics, including *T. sylvaticus*, *S. grimmia*, and *P. monticola*. These findings indicate that physical similarities, including the general morphological size and shape of *T. sylvaticus* and *S. grimmia*, contributed to the probability of correctly classifying them. Similarly, the female bushbuck class and both duiker species, coupled with occlusion led to reduced CCs, particularly for ML methods which may have been due to similar face shapes and markers of both species. This is consistent with previous findings [16,21].

Species size seemed to be a particularly influential characteristic for analysts and ML-DP. The reasoning for this conclusion is that all the species that significantly positively influenced DP were medium-large, except *H. cristata*. However, we removed the size class factor from the analysis due to its high multicollinearity with the factor species. Therefore, although we found a significant correlation between the species and size class factors, we cannot be certain that size was an influencing variable. Yet, there is previous evidence of increasing size positively influencing the DP and CC of wildlife in CT images [26]. Large sizes led to higher DP at greater distances than smaller sizes. Nevertheless, as Gomez Villa et al. [16] reported, species characteristic similarities adversely affect species-specific recognition of similar ungulates when using ML methods. However, human height variance presented no influence on human detection and classification within CT images. Although, as previously stated, much has been studied on wildlife body mass size influences on DP and CC [26], to the best of our knowledge, no study has focused on human height variances and their impact on human DP and CC within CT images. 

The analyst performance factor significantly increased the probability of wildlife CCs. Controversially, Katrak-Adefowora et al. [45] reported low levels of citizen scientist CCs (51.8%) in comparison to professional biologists (77.6%). Significant variation between analyst responses may have been partially due to training methods used prior to analyst-DP and CC testing. 

Darker contrast colours (green) against the background negatively influenced the DP of humans in daylight CT images using the analyst method and CCs using the ML method as opposed to a lighter blue. This is consistent with Hambrecht et al. [2] who found darker contrast colours (green) reduced DP of humans within aerial drone RBG imagery in comparison to lighter colours (red and blue). Additionally, poachers are frequently recorded wearing green and brown camouflage to mix with the background surroundings and reduce being seen [6]. Although much is known about the use of camouflage to disguise oneself, little effort has been focused on the degree of influence colour contrasts have on DP and CC. This study strongly supports that green presents a significantly high degree of influence towards reducing human DP in CT images compared to lighter colours.

The targets orientating in the CTs direction positively influenced human and wildlife DP using the analyst approach and the DP of wildlife using the ML approach. Controversially, Hofmeester et al. [46] found anterior or posterior poses relative to the CT reduced CT trigger probability due to passive infrared (PIR) sensor detection difficulties. However, their study focuses on the influence of target proportions on CT sensor triggers rather than potential influencing target features on target DP in CT images.

In conclusion, distance and occlusion illustrated the most significant influence on both DP and CC, particularly when integrated with dense vegetation for human experiments; vegetation type was not assessed for wildlife experiments. Wildlife CC was particularly influenced by partial or complete occlusion of species, likely caused by occluded visual characteristics. Darker contrast colours significantly reduced human DP. Targets orientating towards the CT illustrated a positive influence on target DP and CC. There was a significant positive variation between analyst performances. Human height presented no influence, and darker lighting conditions significantly negatively influenced human DP and all wildlife models. Additionally, although the analysts performed better overall for the wildlife models, the ML method outperformed analysts for the human experiments (see Table 2 and Table 3). This leads to concluding the proposed hypothesis’ (see Table 4).

### 4.2. Recommendations to Mitigate Influential Factors

The influence of multiple factors on target detection and classification within CT images raises cause for concern if users do not consider these during CT study design and implementation. However, some precautions and actions could be applied, reducing, and potentially mitigating, such influences. 

Many factors should be considered for wildlife monitoring, particularly the monitoring of illegal activity using CTs, including the CT’s structure and appearance (camouflage, robustness, etc.), its setup, and its software capabilities. However, we instead aim to focus on the following CT features, with the aim of directing users towards the most optimal CT type for effective near-real-time use: trigger speed, battery life, field of view, resolution quality, and network capabilities. Additionally, we overview CT distribution and abundance, ML model type and training, analyst training, and Random Subspace Methods (RSM) for managing target occlusions within CT images. 

There are various commercial camera traps available, in addition, most conservation organisations are making strides in advancing CTs, including the PoacherCams Panthera CT (Panthera, New York, NY, USA) [47] and the Trailguard AI camera by RESOLVE (Washington, DC, USA) [48]. Therefore, simply recommending a CT Type will not be of much use due to their ever-evolving technology and the vast range of CTs. Instead, we aim to highlight the factors which we find important for effective near-real-time threat monitoring based on the observations within this study and previous research recommendations [10,49].

The suitability of CTs for various conservation studies is widely acknowledged [50,51,52,53]. However, we focus on CT features that are most important for optimum target detection and classification within CT images. One particularly important feature is trigger speed. This has had a significant influence on target detection rates and has been one of the most evolved aspects of CT features, with trigger speeds now reaching approximately 0.5 seconds [49]. Reliable and long battery life are essential for effective threat monitoring. As previously mentioned most CTs on high sensitivity settings can last, on average, two to four weeks [10]. Additionally, new CT advancements, including Trailguard AI [48], have enhanced rechargeable batteries and can now reach up to 1.5 years on a single battery. More advanced CTs using network power tend to consume more battery, due to the WiFi SD cards drawing power, but advancements in rechargeable batteries and solar energy usage should see improvements with future models [50]. A wide FOV is essential for optimal target detection and classification [54]. The average CT FOV ranges between 40–60° wide and from 5 m up to 30 m depending on the camera height positioning and vegetation density. We also focus on optimal resolution quality, a novel yet important factor for increasing the detection and classification of targets in CT images. Most networked CTs are of high-resolution quality, as standard with 4K video resolution and an average of 30-megapixel images, we recommend such high quality, providing optimum clarity for target detection and classification probabilities. However, one of the most important factors to consider when recommending a CT for near-real-time threat monitoring is the CT’s network capabilities. Commonly, CTs with remote network capabilities rely on cellular connections or WiFi connectivity to send images. However, more recent developments are integrating AI into the CT system to send pre-processed near-real-time information from anywhere to end users [51]. Whytock [53] tested this using a commercial Bushnell Core 24MP Low Glow 119936C CT and customised open-source hardware. The main frame hardware consists of a smart-bridge controller with a custom circuit board, LoRa STM32L0 ultra-low-power microcontroller, and RockBLOCK satellite modem, connecting to a Raspberry Pi 4 Compute module. The system is designed to only consume power when necessary, so it only activates the Raspberry Pi to download and classify images using AI when it receives a message from LoRa (the microcontroller within the smart bridge). After classification of the images, the Raspberry Pi sends the results via satellite and powers down to save battery. This type of open-source method is a reasonably cost-effective option. Users and developers would need to understand the mechanics of integrating and altering open-source hardware, including utilising Raspberry Pi’s for scripting and ML connectivity. However, if users could overcome this issue, this type of CT approach would be highly effective for real-time threat monitoring. 

CT distribution and abundance will vary depending on many conditions, including individual objectives, CT detection range, target range, funding availability, and environmental conditions. However, we simply provide guideline recommendations of CT placement methods for optimum resource use and increased detection probability. 

Based on this paper’s findings we determine that CTs should ideally be placed within 30 m of one another throughout the intended site for optimum detection and classification probability of both humans and wildlife. However, this is not likely feasible for real-world applications. Furthermore, in most cases, this is not required when users apply methods to optimise CT placement. Therefore, we instead offer recommendations for CT placement and distribution methods to increase detection and classification probability. Opportunistic CT placement is the most common approach for long-term wildlife monitoring [55]. The primary focus of CT sites would ideally be within the following landmark areas: feeding and drinking sites, game trails, downed logs, and other minor landmark points, all of which have been found to increase trigger probabilities [55]. However, an alternative method for optimising CT placement methods would be to utilise the spatial monitoring and reporting tool (SMART) as an additional measure to complement optimal CT placement [56]. This tool allows rangers and other users to collaborate and share data including recording wildlife tracking, illegal activity monitoring, poaching camps, traps, and patrol routes, along with current CT and alternative sensor placements. The data can be used to create maps and reports and perform analysis to assess threats effectively and plan for optimal monitoring within specific target areas to prioritise funding and staffing resources. With such implementations, CT sites could be situated at typical hotspots as mentioned above, with a grouping of CTs in each location (depending on the desired area covered) with 15–30 m spacing between them.

For ML applications, we do not go into extensive detail on ML performances and comparisons but instead focus on the most prevalent architectures for target detection and classification as recommended by [57]. Hui [57] compares the Faster R-CNN ResNet101 and Inception ResNet Version 2 (V2), the Region-Based Fully Convolutional Network (R-FCN) ResNet 101, and Single Shot Detector (SSD) with both MobileNet and Inception V2. The R-FCN and SSD architectures are faster overall, with the SSD on MobileNet presenting the highest mAP (19.3) for real-time processing. However, the Faster R-CNN using the Inception ResNet provides the highest accuracy at one FPS for all tested classes. The R-FCN architecture using the Residual Network presents a better balance between speed and accuracy. However, the Faster R-CNN with ResNet 101 can achieve a similar overall performance. Given the overall balance of speed and accuracy, we recommend the Faster R-CNN with ResNet 101 architecture based on performance testing [57] and its tested application within our study. 

The primary recommendation for increasing analyst performance is to apply on-demand resources commonly defined as “just-in-time” training. For example, Katrak-Adefowora et al. [45], who utilised such methods by training 94 citizen scientists (analysts) on wildlife species with CT images, found increased detection rates from 51.8–81.9%. Moreover, after training, analysts reported better confidence in species classifications.

Applying a random subspace method (RSM)—a strategic learning method where the features of a target image are randomly sampled for ML training, has proven successful in managing partial human occlusions. Several studies have successfully detected humans within partially occluded still images, resulting in true-positive detection accuracies of 75.6% [44,58]. Such an application improves target detection performance within partially occluded images without compromising detection accuracy for non-occluded images [44]. This method not only accounts for increased robustness to occlusion but, in turn, potentially reduces the influence of occluded species-specific characteristics on wildlife CC. 

### 4.3. Comparison of ML and Analyst Methods for DP and CC Performance

In this study, the overall wildlife classification and detection rates using the ML approach were low compared to similar studies, including detection and classification estimates of up to 93% probability [22]. Such low rates are potentially due to a low (1000 count) threshold of training images per class. Similar studies report dataset counts ranging from 22,000 [16] to 189,000 [19]. Moreover, Gomez Villa et al. [16] report that model training influences wildlife DP. However, the ML approach demonstrated increased DPs within daylight and dusk human DP models. Moreover, for the dusk DP experiments, the ML approach performed better than the analyst approach. However, low analyst-CC results may be partially contributed to by analyst experience.

## 5. Conclusions

This study demonstrated that the ML method displayed significantly better detection probabilities for human daylight and dusk models compared to the analyst method. However, the ML method illustrated less effective detection and classification probabilities for wildlife models than the analyst method. On the other hand, the analyst method performed well with high detection probabilities of wildlife and humans within CT images, with negligible significance on wildlife CC. Overall the ML method showed great promise for future applications. Therefore, despite such performances for wildlife models, we highly recommend the ML method for target detection and classification within CT images if users carefully consider model training and preparation. Regarding analyst approaches, the developed knowledge of citizen scientist-based factors and influences on wildlife DP and CC could increase the accuracy of ecological monitoring evaluations. Potential improvements to increase DPs and CCs include increased training datasets per class, with a minimum of 2000 per class, considering characteristically similar species. Moreover, future considerations include categorising wildlife into size classes by body mass measurements rather than average height at shoulder length. This includes additional measures for with and without partially occluded species characteristics and directly comparing similar species. Additionally, we recommend providing further training to citizen scientists using “just-in-time” methods prior to image analysis.

## Figures and Tables

**Figure 2 sensors-22-05386-f002:**
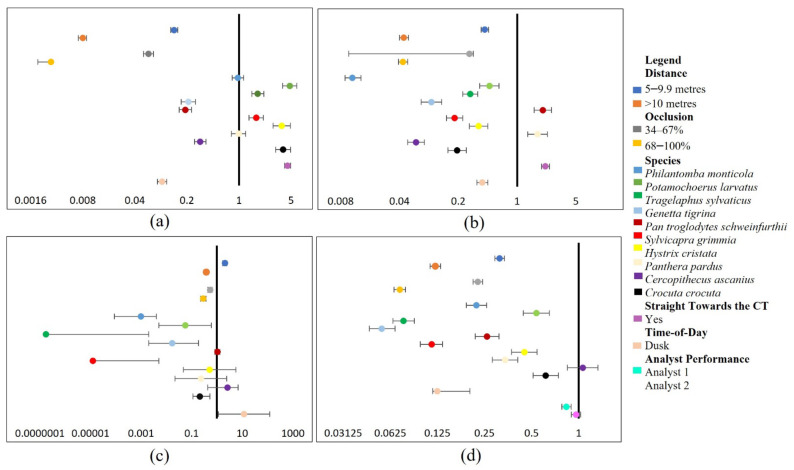
The best-fit models for the wildlife experiments presented as forest plots. All forest plots are composed of effect sizes and error bars containing 95% confidence intervals. Moreover, each factor class is compared to a baseline class as follows, distance: 0–4.9 m, occlusion: 0–33%, species: *Papio cynocephalus*, orientation towards the CT: no, analyst performance: Analyst 1, and time-of-day: daylight. The plots are illustrated as follows: (**a**) analyst detection probability (DP) model; (**b**) machine learning (ML) DP model; (**c**) ML correct classification (CC) model; (**d**) analyst-CC model.

**Figure 3 sensors-22-05386-f003:**
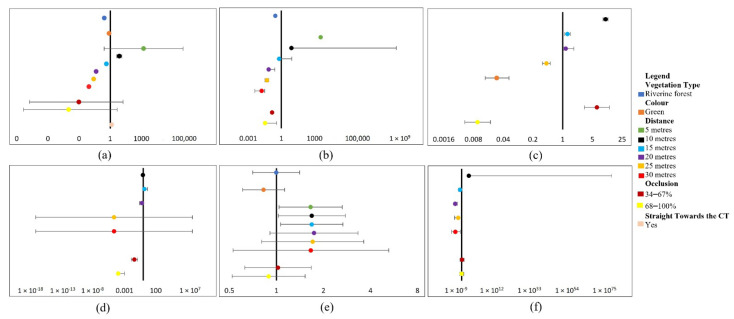
The best-fit models for the human experiments presented as forest plots. All plots are composed of effect estimates and error bars containing 95% confidence intervals. Additionally, all factor categories are compared with a reference category which are as follows: distance (daylight experiments): <5 m, distance (dusk experiments): 5 m, occlusion: 0–33%, vegetation type: miombo woodland, colour: blue, and orientation towards the CT: no. The plots are illustrated as follows: (**a**) analyst-daylight-DP model; (**b**) ML-daylight-DP model; (**c**) analyst-dusk-DP model; (**d**) ML-dusk-DP model; (**e**) ML-daylight-CC model; (**f**) ML-dusk-CC model.

**Table 1 sensors-22-05386-t001:** The tested variables within both human and wildlife experiments, showing each variable name, their type of effect on the experimental outcome, their associated classification (dummy data) label, and each variable type.

Variable	Effect	Classification Labels	Variable Type
Outcome Variable			
Detection ^1^	Target	False-negative = 1/True-positive = 2	Binary
Correct Classification ^1^	Target	False-negative = 1/True-positive = 2	Binary
Predictor Variables			
Occlusion ^1^	Fixed	0–33% = 1, 34–67% = 2, and 68–100% = 3	Nominal
Distance ^1^	Fixed	<5, 5, 10, 15, 20, 25, and 30 m/0–4.9 = 1, 5–9.9 = 2, and ≥10 = 3	Nominal
Orientation Towards the CT ^1^	Fixed	Yes = 1/No = 2	Binary
Analyst Performance ^1^	Fixed	Analyst 1, 2, 3	Nominal
Vegetation Type ^2^	Fixed	Miombo Woodland = 1/Riverine Forest = 2	Binary
Colour ^2^	Fixed	Blue = 1/Green = 2	Binary
Height ^2^	Fixed	Participant height in metres	Continuous
Species ^3^	Fixed	Species scientific name identified numerically (1–11)	Nominal
Size Classes ^3^	Fixed	Small = 1, Medium = 2, and Large = 3	Nominal
Hierarchical Variables			
Experiment Number ^2^	Random	Identification number of the experiment	Nominal
Tree Tag Number ^3^	Random	Identification number of the tree tag location	Nominal

Notes: ^1^ both wildlife and human factors; ^2^ human factors; ^3^ wildlife factors.

**Table 2 sensors-22-05386-t002:** The percentage of true-positive detections and classifications compared to true-positive detections, false-negative classifications, and false-negative detections and no classifications for machine learning (ML) (3697, 3340, 9457) and analyst (30,436, 9279, 9281) methods.

Model	True-Positive Detection and Classification %	True-Positive Detection and False-Negative Classification %	False-Negative Detection and No Classification %
Machine Learning	22.4	20.3	57.3
Analyst	62.1	18.9	19.0

**Table 3 sensors-22-05386-t003:** The percentage of true-positive detections and classifications vs. false-negative detections and no classifications for ML daylight (8474 vs. 2590), ML dusk (19,842 vs. 16,343), analyst daylight (18,250 vs. 7172), and analyst dusk (9634 vs. 49,893) for all human models. We found no false-negative classification outcomes for human models.

Model	True-Positive Detection and Classification %	False-Negative Detection and No Classification %
ML daylight	76.5	23.4
ML dusk	54.8	45.1
Analyst daylight	71.7	28.2
Analyst dusk	16.1	83.8

**Table 4 sensors-22-05386-t004:** The tested hypothesis summary for all factors.

Alternative Hypothesis	Outcome
Targets orientating towards the CT would positively influence target DP in CT images.	Accepted
Targets orientating towards the CT would positively influence target CC in CT images.	Rejected
Partially occluded species-specific characteristics would negatively affect CC of similar species.	Accepted
The time-of-day factor (dusk) would negatively influence wildlife DP and CC.	Accepted
The time-of-day factor (dusk) would negatively influence human DP and CC.	Rejected
Colour contrast (green), in comparison to the background, would negatively impact human DP.	Accepted
Increasing distance and occlusion would significantly decrease target DP and CC for all wildlife models.	Accepted
Increasing distance and occlusion would significantly decrease target DP and CC for all human models.	Rejected
Dense vegetation would contribute to the significant decrease of one or more models.	Accepted
ML methods could perform at a statistically significant increased rate than analyst methods for at least one model.	Accepted
There would be a significant positive difference in analyst performance on target DP and CC.	Accepted
Decreases in size would decrease DP and CC, respectively, for wildlife and humans.	Rejected

## Data Availability

The data presented in this study are available in Supplementary Spreadsheets S1–S10.

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
