# Peer review of "Understanding External Influences on Target Detection and Classification Using Camera Trap Images and Machine Learning"

_sensors, 2022, doi:10.3390/s22145386_

Round 1

Reviewer 1 Report

1. The authors use ResNet model for ML experiments. As you know, ResNet is a complex neural network with millions of trainable parameters. Do you use network pruning or weight quantization techniques to reduce the network size while maintaining the same/similar detection accuracy? Such as

[1] Huang, Q. Weight-Quantized SqueezeNet for Resource-Constrained Robot Vacuums for Indoor Obstacle Classification. AI 2022, 3, 180-193. https://doi.org/10.3390/ai3010011

[2] Z. Tang, et al., Automatic Sparse Connectivity Learning for Neural Networks, IEEE Transactions on Neural Networks and Learning Systems, 2022.

2. Transfer learning technique is also adopted in the experiment. Please report more details of how to do transfer learning. Do you use the recent transfer learning techniques, such as

[3] J. Zheng, C. Lu, C. Hao, D. Chen and D. Guo, "Improving the Generalization Ability of Deep Neural Networks for Cross-Domain Visual Recognition," in IEEE Transactions on Cognitive and Developmental Systems, vol. 13, no. 3, pp. 607-620, Sept. 2021, doi: 10.1109/TCDS.2020.2965166.

[4] Cong Hao, Deming Chen, "Software/Hardware Co-design for Multi-modal Multi-task Learning in Autonomous Systems", 2021 IEEE 3rd International Conference on Artificial Intelligence Circuits and Systems (AICAS), pp.1-5, 2021.

Author Response

Dear Editors and Reviewer,

We have made the amendments to the manuscript as according to the comments made by each reviewer. The amendments made or answers given will be detailed below alongside the corresponding reviewer comments or questions.

Question 1: The authors use ResNet model for ML experiments. As you know, ResNet is a complex neural network with millions of trainable parameters. Do you use network pruning or weight quantization techniques to reduce the network size while maintaining the same/similar detection accuracy?

Answer: We don’t use weight pruning or quantization, because the GPUs were running the models on are high performance models so there is no need.

Question 2: Transfer learning technique is also adopted in the experiment. Please report more details of how to do transfer learning. Do you use the recent transfer learning techniques?

Answer: We did use transfer learning. Here we use the frozen weights of the ResNet101 and retrain the fully connected multilayer perceptron (MLP). We have included more information regarding the transfer learning method used (found on lines 296-300).

This concludes my amendments as according to reviewer requests. I do hope such information is sufficient and I look forward to hearing your responses in the near future.

Reviewer 2 Report

This paper presents Understanding External Influences on Target Detection and Classification using Camera Trap Images and Machine Learning.

The work looks interesting, but the experimental part has to improve prior to the acceptance. Comments to revise the manuscript are as follows; 1. The discussion regarding the merits of CT image usage is not enough. Authors have to make this discussion in a more scientific manner, emphasizing features of  CT images and how those features are good to carry on this work. 2. In the theory part, there are no equations to present the contents regarding the theories.  So, I recommend improving the theory part in a more comprehensive manner. 3.The performance of the experiments has to be improved by analyzing based on the confusion matrices.  4. In Table 2, If you just say "ML" it would be difficult for readers to follow your work. I recommend you to mention name of the model instead of the "ML"

Author Response

Dear Editors and Reviewer,

We have made the amendments to the manuscript as according to the comments made by each reviewer. The amendments made or answers given will be detailed below alongside the corresponding reviewer comments or questions.

Comment 1: The discussion regarding the merits of CT image usage is not enough. Authors must make this discussion in a more scientific manner, emphasizing features of CT images and how those features are good to carry on this work.

Amendments: I have more comprehensively discussed the advantages of using camera traps as opposed to comparable tools and emphasised the features which make this tool a recommended approach for this work (found on lines 41-61). In addition, I have overviewed these factors in the discussion and used those as basis for camera trap type recommendations (found on lines 652-701).

Comment 2: In the theory part, there are no equations to present the contents regarding the theories.  So, I recommend improving the theory part in a more comprehensive manner.

Amendments: I was not entirely certain on what it is that the reviewer was requesting. However, I have tried to be more comprehensive of my theory based on the data that was presented and I have made an effort to include equations of my theories to help visually summarise the main points (lines 529-548).

Comment 3: The performance of the experiments must be improved by analysing based on the confusion matrices.

Amendments: I have calculated and included a multi-class confusion matrix for all classes within the model. Additionally, I have included a table of the F1 score, precision, recall, false-negative rate, false-positive rate, and true negative rate for each class. This is found in the Appendix C section, Table’s C1 and C2.

Comment 4: In Table 2, If you just say "ML" it would be difficult for readers to follow your work. I recommend you mention name of the model instead of the "ML"

Amendments: This has been changed to the full word, found within Table 2.

This concludes my amendments as according to reviewer requests. I do hope such information is sufficient and I look forward to hearing your responses in the near future.

Reviewer 3 Report

The paper analyzes the influence factors for ML based wildlife detection and recognition. According to the experimental comparison, the paper concludes that detection should prefer to ML while classification should prefer to human analysis. Generaly speaking , the paper has some interesting contributions for wildlife survey.

Author Response

Thanks for the positive comments about the paper.

Round 2

Reviewer 2 Report

N/A